# No learning rates needed: Introducing SALSA - Stable Armijo Line Search Adaptation

## Abstract

In recent studies, line search methods have been demonstrated to significantly enhance the performance of conventional stochastic gradient descent techniques across various datasets and architectures, while making an otherwise critical choice of learning rate schedule superfluous (Vaswani et al., 2019; Mahsereci & Hennig, 2015; Vaswani et al., 2021). In this paper, we identify problems of current state-of-the-art of line search methods (Vaswani et al., 2019; 2021), propose enhancements, and rigorously assess their effectiveness. Furthermore, we evaluate these methods on orders of magnitude larger datasets and more complex data domains than previously done.

More specifically, we enhance the Armijo line search method by speeding up its computation and incorporating a momentum term into the Armijo criterion, making it better suited for stochastic mini-batching. Our optimization approach outperforms both the previous Armijo implementation and a tuned learning rate schedule for the Adam and SGD optimizers. Our evaluation covers a diverse range of architectures, such as Transformers, CNNs, and MLPs, as well as data domains, including NLP and image data.

Our work is publicly available as a Python package, which provides a hyperparameter free Pytorch optimizer.

## 1 Introduction

In the field of modern machine learning, there are numerous optimization algorithms available (Schmidt et al., 2021). However, determining the most suitable algorithm for a specific problem and finding the appropriate learning rate or learning rate schedule often requires extensive expertise and computational resources. In particular, the prevailing approach involves treating the learning rate as a hyperparameter and training the network repeatedly until the optimal value that yields the best performance is discovered. To simplify and expedite this process, recent research in deep learning (Vaswani et al., 2019; Mahsereci & Hennig, 2015; Bollapragada et al., 2018; Paquette & Scheinberg, 2020) has proposed the reintroduction of line search methods as popular optimization technology, which effectively identify an adaptive learning rate by evaluating the loss function at different points along the gradient direction, thus eliminating costly hyperparameter tuning.

As traditional line search requires multiple forward passes per gradient update, a more efficient approach is desirable. In Vaswani et al. (2019), a Stochastic Line Search (SLS) has been combined with a smart re-initialization of the step size to alleviate the need for multiple forward passes for every step. This approach was shown in Vaswani et al. (2019) to improve a variety of optimization methods, such as Stochastic Gradient Descent (SGD) on tasks such as matrix factorization as well as image classification for small networks and datasets. In Vaswani et al. (2021) the authors adapt this line search to preconditioned optimizers like Adam (Kingma & Ba, 2015) further increasing its usability.

In this paper we extend upon this work, by introducing a momentum term to the SLS, critically improving its performance and stability. Furthermore, we introduce a limitation on the frequency with which a line search is performed, greatly reducing the computation needed. Additionally, we conduct extensive experiments to evaluate the performance of various optimization methods across different datasets domains and architecture options. Our findings demonstrate that, our improved Stable Armijo Line Search Adaptation algorithm, called SaLSa, consistently outperforms the previously introduced SLS as well as tuned optimizers, with very little computational overhead (about 3% compared to no

line search). We observe the SaLSa optimizers have on average an 1.5% advantage on accuracy and a 50% lower average log loss at end of training. Additionally, the stability of the training is improved compared to previously introduced versions of SLS.

To make our work easy to reproduce and use, we implement all methods as PyTorch optimizers. The source code is open-source and free software (MIT licensed) and available at [anonymized URL for review. Code is available in the supplementary material]

## 2 BACKGROUND

The stochastic Armijo line search described in Vaswani et al. (2019) is designed to set a step size for all network parameters $w_k$ at iteration $k$. In this section, we formulate a modification of the Armijo criterion to handle the ADAM (Kingma & Ba, 2015) direction instead of the classical SGD direction. This is based upon Vaswani et al. (2019; 2021) Moreover, we introduce an improved Armijo criterion, which mitigates the effect of noise in the mini-batch setting by calculating an exponential moving average smoothing on both sides of the Armijo equation.

We define the following notation: The loss function is denoted by $f(w)$. $|| \cdot ||$ denotes the Euclidean norm and $\nabla f$ denotes the gradient of $f$. Given the iteration counter $k$, $f_k$ and $\nabla f_k$ denote the mini-batch loss and its mini-batch gradient.

### 2.1 ARMIJO LINE SEARCH

The Armijo line search criterion is defined in Vaswani et al. (2019) as:

$$f_k(w_k + \eta_k d_k) \leq f_k(w_k) - c \cdot \eta_k ||\nabla f_k(w_k)||^2, \tag{1}$$

where $d_k$ is the direction (e.g., $d_k = -\nabla f_k(w_k)$ in case of SGD), $c \in (0, 1)$ is a constant (commonly fixed to be $0.1$ Vaswani et al. (2019)). The step size $\eta_k$ which satisfies Condition 1 is practically obtained by employing a backtracking procedure, i.e., starting with a high initial step-size $\eta_k^0$ and iteratively decreasing it by a constant factor $\delta \in (0, 1)$ until Condition 1 is satisfied (in practice $\delta = 0.9$).

To avoid a monotonically decreasing step size, $\eta_k$ is increased each step by the following formula:

$$\eta_k^0 = \eta_{k-1} \cdot 2^{1/b} \tag{2}$$

as described in Vaswani et al. (2019). In practice for $b = 500$, this will usually avoid backtracking multiple times per step, since the increase in step size is small. Henceforth, we will refer to this algorithm as SLS.

### 2.2 INCLUDING ADAM'S UPDATE STEP IN SLS

In case of SGD, the direction $d_k$ is the negative mini-batch gradient.

$$d_k = -\nabla f_k(w_k)$$

Adam's direction defined in Kingma & Ba (2015) can be written as:

$$\begin{aligned}
g_k &= \nabla f_{ik}(w_k) \\
m_k &= \beta_1 \cdot m_{k-1} + (1 - \beta_1) \cdot g_k \\
v_k &= \beta_2 \cdot v_{k-1} + (1 - \beta_2) \cdot g_k^2 \\
\hat{m}_k &= m_{k-1}/(1 - \beta_1^k) \\
\hat{v}_k &= v_{k-1}/(1 - \beta_2^k) \\
d_k &= -\hat{m}_k/(\sqrt{\hat{v}_k} + \epsilon)
\end{aligned} \tag{3}$$

Adam combines a momentum-based approach together with a step size correction built upon

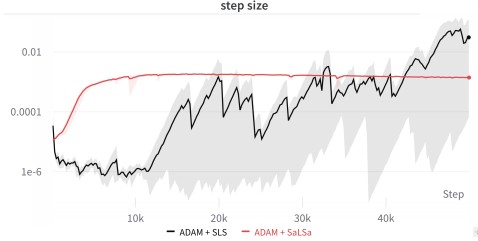

Figure 1: The step size of ADAM + SLS as well as ADAM + SaLSa on ImageNet. Colored areas indicate variance between runs. Notice the large variations for ADAM + SLS compared to the consistent and stable behavior of ADAM + SaLSa.

the gradients variance. In the training of Transformers, these modifications have been shown to be important enhancements over the simpler SGD algorithm (Kunstner et al., 2023). The weight update rule is generally defined as

$$w_{k+1} = w_k + \eta_k d_k. \tag{4}$$

The Armijo line search criterion from Eq. 1 must be adjusted for the Adam optimizer. We perform this adjustment based on Vaswani et al. (2019; 2021). To check if the Armijo line search criterion is satisfied in the Adam case, we use the direction $d_k$ defined in Eq. 3, with momentum $\beta_1 = 0$. Note that, the Armijo criterion is only guaranteed to be satisfy-able by adjusting the step size $\eta_k$, if the update direction and the gradient direction are identical. However, this condition is not met when $\beta_1 \neq 0$ in Eq. 3. Additionally, we replace the gradient norm term $||\nabla f_k(w_k)||^2$ by the preconditioned gradient norm $\frac{||\nabla f_k(w_k)||^2}{\sqrt{\hat{v}_k} + \epsilon}$ as in Vaswani et al. (2021) resulting in Eq. 5.

$$f_k(w_k + \eta_k d_k) \leq f_k(w_k) - c \cdot \eta_k \frac{||\nabla f_k(w_k)||^2}{\sqrt{\hat{v}_k} + \epsilon} \tag{5}$$

Note that to perform final weight updates each step we use $\beta_1 \neq 0$.

### 2.3 SLS FAILURE CASES

As shown in Vaswani et al. (2019; 2021) the previously described line search methods perform well on smaller datasets and neural network architectures. However, here we show that these methods have problems to consistently perform during larger scale training.

The first of these Problems we call "mini-batch noise": Eq. 1 and 5 describe criterions which are checked for every mini-batch. This is problematic, since the criteria will be violated subject to inherent noise in the mini-batch data. The phenomenon is amplified by small mini-batch sizes. As can be seen in Figure 2 in a typical training run the Armijo line search method leads to frequent changes of the step size, see Figure 2.

Another frequently occuring Problem is mini-batch gradient approximations $\nabla f_k(w_k) \approx 0$. These are due to computational precision problems, even with float32 precision enabled. For an example see Figure 3. In the original implementation by Vaswani et al. (2021), whenever $\nabla f_k(w_k) \leq 10^{-8}$ no line search was performed.

Additionally, a problem which only occurs on large datasets, can be seen in Figure 1. The step size and its variance over 5 different runs on ImageNet is visualized. We observe that the step sizes of SLS are very sensitive to initial conditions, the only difference between the runs is the random parameter initialization of the network and the shuffled dataset. This problem

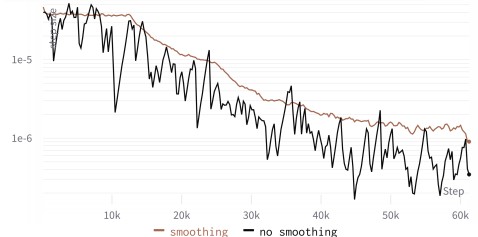

Figure 2: The step size of ADAM + SLS (black) compared to ADAM + SaLSa (brown) visualized during a training run of BERT on the MNLI dataset.

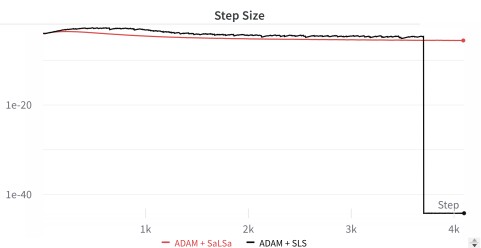

Figure 3: The step size of ADAM + SLS (black) on the CIFAR10 dataset. We sometimes observed drastic drops in step size due to computational precision problems. When training with SaLSa (red) we do not observe any such drops.

only seems to occur on larger datasets and is quite impactful, as some runs do not converge properly, or take very long to even begin converging.

## 3 METHODS

To obtain a line search method with better properties in the mini-batch scenario, we propose to extend the Armijo criterion with a momentum term. Below we provide a detailed explanation of the modifications we made, the theoretical basis behind them, and our reasoning. Furthermore, we introduce a method to greatly reduce the computational overhead introduced by a line search.

### 3.1 ADDRESSING MINI-BATCH NOISE

As an extension to Eq. 1 we propose an exponential smoothing (also called momentum term) of all factors which are dependent on the mini batch in this equation. First we rewrite Eq. 1:

$$f_k(w_k) - f_k(w_k + \eta_k d_k) \geq c \cdot \eta_k ||\nabla f_k(w_k)||^2 \tag{6}$$

$f_k(w_k) - f_k(w_k + \eta_k d_k)$ denotes the decrease in loss and $||\nabla f_k(w_k)||^2$ denotes the gradient norm. In order to apply exponential smoothing to both terms we define $h_k$ and $s_k$ as follows:

$$h_k = h_{k-1} \cdot \beta_3 + (f_k(w_k) - f_k(w_k + \eta_k d_k)) \cdot (1 - \beta_3)$$
$$s_k = s_{k-1} \cdot \beta_3 + ||\nabla f_k(w_k)||^2 \cdot (1 - \beta_3) \tag{7}$$

$h_k$ represents the smoothed decrease of the loss with the current step size, $s_k$ the smoothed gradient norm and $\beta_3 \in (0, 1)$ the smoothing factor used for the exponential moving average.

We introduce the Stable Armijo Line Search Adaptation (SaLSa) criterion as:

$$h_k \geq c \cdot \eta_k \cdot s_k \tag{8}$$

Combining SaLSa and the Adam optimizer is done by computing $s_k$ as follows:

$$s_k = s_{k-1} \cdot \beta_3 + \frac{||\nabla f_k(w_k)||^2}{\sqrt{\hat{v}_k} + \epsilon} \cdot (1 - \beta_3) \tag{9}$$

and computing $d_k$ as described in Equation 3, but with $\beta_1 = 0$. We keep the calculation procedure for the step size $\eta_k$ the same as previously described.

### 3.2 INTUITIVE MOTIVATION

As mentioned in Section 3 due to the inherent noise in mini-batches we expect some of them to violate the original Armijo line search, even if the step size $\eta_k$ is appropriate for the majority of mini-batches around step $k$ in training.

Let us assume that all mini-batches are normally distributed with respect to the Armijo criterion, e.g. some mini-batches fulfill the condition with a wide margin, most fulfill it with a small margin and some rare exceptional batches violate the criterion. In this scenario we do not want to decrease the step size by a large amount each time we get an exceptionally bad mini-batch, since the step size is still fitting for most batches and Equation 2 is only increasing the step size slowly. The stable Armijo line search adaptation in Equation 8 is implementing exactly this behaviour. Note that the exact distribution is not relevant in this thought experiment.

If we analyze Equation 8, we notice that the right hand side is affected by the current step size $\eta_k$ to the same degree as in the original Armijo line search Equation1. However, the left hand side is substantially less affected since $\eta_k$ is part of the exponential smoothing process. This results in a slower reduction of the step size $\eta_k$ as the criterion is fulfilled more easily than the original criterion by lowering $\eta_k$.

### 3.3 THEORETICAL ANALYSIS

We extend the convergence Theorem introduced in the original Armijo paper (Armijo, 1966) for the full batch setting and the SaLSa criterion with SGD from Eq.8. We additionally require that

every found learning rate yields an improved loss $f(w_k) - f(w_k + \eta_k d_k) \geq 0$. This condition ensures the convergence for an infinite sequence, which may otherwise not be guaranteed due to the exponential smoothing. In practice, enforcing this condition did not yield a significant difference in the optimization process and is thus not implemented in our experiments. The proof for the theorem below, as well as training runs with this condition, can be found in the Appendix A.3.

**Theorem 1** (Convergence Theorem). *Let $f \equiv f_k$. For $w_0 \in \mathbb{R}^d$ let $S(w_0) := \{w \mid f(w) \leq f(w_0)\}$ and assume that $f(w^*) := \inf_{w \in \mathbb{R}^d} f(w)$ exists for a unique point $w^* \in \mathbb{R}^d$ with $\nabla f(w) = 0$ for $w \in S(w_0)$ if and only if $w = w^*$. Any sequence $\{w_k\}_{k=1}^{\infty}$ found by the SaLSa criterion with $f(w_k) - f(w_k + \eta_k d_k) \geq 0$ and $c < 1$ converges to $w^*$.*

## 3.4 ADDRESSING COMPUTATIONAL COSTS

It is unnecessary and computationally expensive to perform a line search for every step during training, as for most steps the step size does not need to be changed. The overall training compute cost increases by roughly 30% when performing a line search every step. To address this, we propose to perform a line search more regularly when a high rate of change of the step size is detected and less regularly otherwise. We realize this by keeping 2 different exponential moving averages of the step size $\eta_k$ which we update after every line search procedure:

$$\bar{\eta}_k(\beta) = \beta \bar{\eta}_{k-1} + (1 - \beta) \cdot \eta_{k-1} \tag{10}$$

We calculate the average rate of change as follows:

$$r_k = \frac{\bar{\eta}_k(0.9)}{\bar{\eta}_k(0.99)} \tag{11}$$

and invert it if $r_k \leq 1$:

$$\bar{r}_k = \begin{cases} r_k & \text{if } r_k \geq 1 \\ r_k^{-1} & \text{otherwise} \end{cases} \tag{12}$$

we set the line search frequency $L_k$ to the closest integer of:

$$L_k = \frac{1}{\bar{r}_k - 1} \tag{13}$$

and clamp it to the range $L_{k+1} \in [1, 10]$. We perform the line search every $L_{k+1}$ steps. This reduces the extra compute needed from roughly 30% to approximately 3% for longer runs. In practice, we did not notice any performance degradation, see the Appendix for ablation studies A.5.

## 3.5 PRACTICAL CONSIDERATIONS

In the original Armijo line search implementation a few outliers dominated the determination of $\eta_k$ as shown in Figure 2. The hyperparameter $c$ was set with this in mind. In our experiments we found good values for $c$ to be in the range $c \in [0.3, 0.5]$. For all our experiments we used $c = 0.3$ (compared to $c = 0.1$ for the original Armijo line search criterion). Ablation studies on the impact of different $c$ values can be found in the Appendix.

Furthermore, we tuned the hyperparameter $\beta_3 \in [0.9, 0.999]$ from Eq. 7 on a variety of datasets. We found that although performance is robust to the choice of $\beta_3$, a value of $\beta_3 = 0.99$ is the best general choice. Larger $\beta_3$'s result in slower adaptation of the step size to the loss landscape, but less susceptibility to noise. Ablation studies on the impact of different $\beta_3$ values can be found in the Appendix A.4.

## 4 EXPERIMENTAL APPROACH

In this section, we detail our experimental design to investigate the performance of our proposed optimization method. We utilize datasets, model implementations and weights from the Huggingface library (Wolf et al., 2019), the pytorch datasets library and the nanoGPT (Karpathy, 2023) github repository.

## 4.1 CANDIDATES

A quick overview of all candidates we are evaluating can be seen below:

- SGD with tuned learning rate and learning rate schedule

- ADAM with tuned learning rate and learning rate schedule

- SGD + SLS, see Section 2.1

- ADAM + SLS, see Section 2.2

- SGD + SaLSa, see Section 3

- ADAM + SaLSa, see Section 3

As a baseline comparison we evaluate the ADAM and SGD optimizers with a cosine decay with warm starting for 10% of the total training time. For NLP tasks this warm starting and cosine decay is common practice. For the image tasks we compare to a flat learning rate as done in Vaswani et al. (2019).

We take the peak learning rate for ADAM on natural language tasks $\eta = 2 \cdot 10^{-5}$ from the original Bert paper by Devlin et al. (2019), which presents a good value for a variety of classification tasks, including the Glue (Wang et al., 2018) tasks upon which we are evaluating. We found the value for the peak learning rate for SGD on the NLP task $\eta = 2 \cdot 10^{-3}$ using a grid search.

We found the value $\eta = 1 \cdot 10^{-3}$ for image classification for ADAM using a grid search. The same procedure resulted in $\eta = 1 \cdot 10^{-1}$ for SGD.

## 4.2 DATASETS AND MODELS

To evaluate an optimization method it is necessary to perform large scale runs of complex real world datasets and tasks. This is especially important as many optimization methods perform well on small scale or clean theoretical tasks, but fail to perform well on real world data.

*Natural Language Processing - Transformers*

We consider a common scenario in natural language processing, where a large pre-trained language model (in our case Bert (Devlin et al., 2019)) is fine-tuned on a small to medium sized dataset. The Glue dataset by Wang et al. (2018) is a collection of various popular classification tasks in NLP, and it is widely used to evaluate common natural language processing capabilities. All datasets used are the version provided by tensorflow-datasets 4.0.1.

More specifically of the Glue collection (Wang et al., 2018), we use the datasets Stanford Sentiment Treebank *SST2*, Microsoft Research Paraphrase Corpus *MRPC*, Stanford Question Answering Dataset *QNLI*, and the Multi-Genre Natural Language Inference Corpus *MNLI*. These datasets range from 500 - 400.000 training samples and represent a variety of fine-tuning tasks.

As a further evaluation metric for language models, we fine-tune GPT-2 (Radford et al., 2019) on the Shakespeare dataset as described in Karpathy (2023) implementation.

*Image Classification - Convolutional Neural Networks*

In image classification common evaluation datasets are CIFAR10 and CIFAR100 (Krizhevsky, 2009), both being small scale (50.000 samples, 32x32 resolution). To obtain more reliable results we also compare on ImageNet (Deng et al., 2009) which consists of roughly $10^6$ samples. We use the ResNet34 (He et al., 2016) architecture without pre-training for small datasets and ResNet50 for ImageNet.

## 4.3 IMPLEMENTATION DETAILS

The following details are the same for all experiments: All models are trained 5 times and the averaged metrics are reported in Tables 1 and 2. The learning curves as well as standard errors are visualized in Figures 4 and 5.

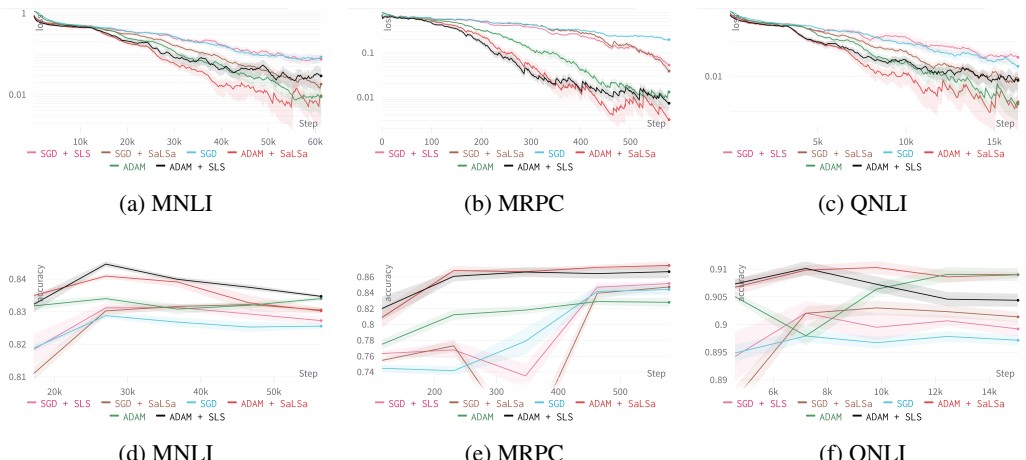

Figure 4: The loss (top row) and accuracy curves (bottom row) of the experiments on the GLUE dataset with standard error indicated around each line, starting after the first epoch. Accuracy was calculated on the validation data, while loss was calculated on the training data.

A Bert Devlin et al. (2019) model was trained on the NLP dataset with the following hyperparameter choices: Five epochs training time on each dataset. The pooling operation used in the Glue experiments is [CLS]. The maximum sequence length is set to 256 tokens. The batch size used during training is 32.

For the image datasets CIFAR10 and CIFAR100 (Krizhevsky, 2009) ResNet34 (He et al., 2016) was used. For the ImageNet (Deng et al., 2009) dataset the ResNet50 (He et al., 2016) architecture was used, a larger architecture was used due to the increased amount of complexity and size of the dataset. The batch size used during training is set to 128 for Cifar10 and Cifar100 and 256 for ImageNet. We applied pre-processing as described in the ResNet paper by He et al. (2016). Models were trained on CIFAR10 and CIFAR100 for 100 epochs and on ImageNet for 12 epochs.

The computing time for all experiments was roughly 65 days on an A40 GPU. Roughly 15 of these days were used for the NLP tasks and 50 for the image datasets.

## 5 Experimental Results

In this section, we will describe the results of our experiments. We compare the 6 candidates as described in Section 4.1. All metrics reported are average values obtained using 5 training runs. All accuracies displayed are calculated on the validation sets. The losses displayed are calculated on the training sets, smoothed with exponential moving average. The colored areas around each line indicate the standard error of each experiment. We display the accuracies and losses during the training period in Figures 4 and 5. The Tables 1 and 2 are displaying peak accuracies and final loss of each candidate. Additional visualizations of experimental results can be found in the Appendix A.2. In summary, we observe the SaLSa methods having on average an 1.5% advantage on accuracy and a 50% lower average log loss at end of training.

### 5.1 Natural Language Processing - Transformer Experiments

In our NLP experiments, as shown in Figure 4 and in the Appendix for GPT-2 and SST2, we have observed that, on average, ADAM + SaLSa achieves a lower final loss compared to ADAM, ADAM + SLS, and SGD + SLS. However, this improvement in loss does not always translate to a significant difference in the accuracy metric. ADAM + SLS and ADAM + SaLSa perform similarly in terms of accuracy, but both outperform ADAM and SGD + SLS on average. Note that for similar final losses, the convergence rate of SaLSa is generally faster than that of ADAM, as depicted in Figure 4.

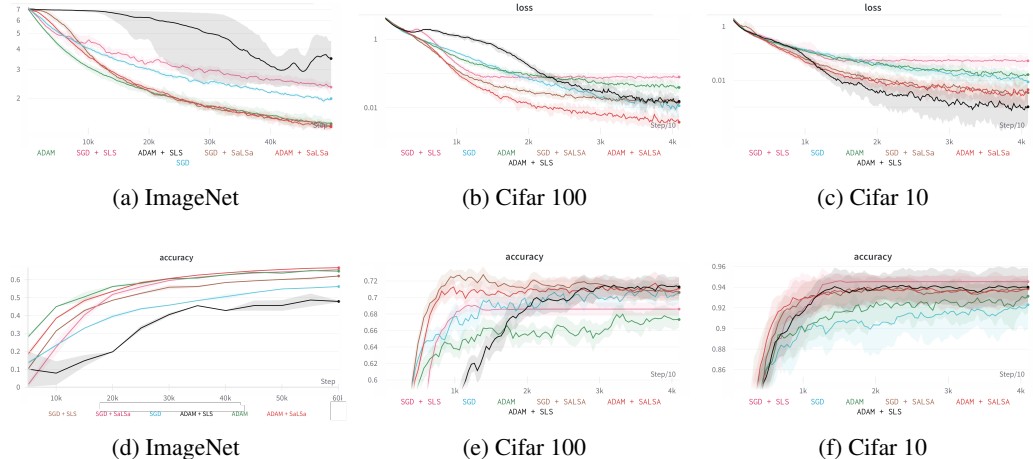

(a) ImageNet  (b) Cifar 100  (c) Cifar 10

(d) ImageNet  (e) Cifar 100  (f) Cifar 10

Figure 5: The loss (top row) and accuracy curves (bottom row) of the ResNet experiments on the image datasets with standard error indicated around each line, starting after the first epoch. Accuracy was calculated on the validation data, while loss was calculated on the training data.

Table 1: Peak classification accuracies, averaged over 5 runs, for all datasets and optimization methods. Best performing optimization method is marked in **bold**.

|  | ADAM - | SGD - | ADAM SLS | SGD SLS | ADAM SaLSa | SGD SaLSa |
|---|---|---|---|---|---|---|
| *MNLI* | 0.8340 | 0.8256 | **0.8446** | 0.8303 | 0.8409 | 0.8305 |
| *QNLI* | **0.9090** | 0.8972 | 0.9044 | 0.8971 | **0.9090** | 0.9014 |
| *MRPC* | 0.8279 | 0.8441 | 0.8667 | 0.8441 | **0.8745** | 0.8473 |
| *SST2* | **0.9271** | 0.9225 | 0.9261 | 0.9167 | 0.9216 | 0.9245 |
| ResNet34 |  |  |  |  |  |  |
| *CIFAR10* | 0.9312 | 0.9229 | 0.9401 | **0.9453** | 0.9389 | 0.9384 |
| *CIFAR100* | 0.6733 | 0.7057 | **0.7128** | 0.6859 | 0.7114 | 0.7064 |
| ResNet50 |  |  |  |  |  |  |
| *ImageNet* | 0.6486 | 0.5633 | 0.4836 | 0.6234 | **0.6684** | 0.6594 |
| average | 0.8215 | 0.8116 | 0.8111 | 0.8204 | **0.8378** | 0.8297 |
| average rank | 3.57 | 4.86 | 2.42 | 4.43 | **2.14** | 3.28 |

## 5.2 IMAGE - CONVOLUTIONAL NEURAL NETWORKS EXPERIMENTS

In our image experiments, we have observed that the combination of ADAM + SLS or SGD + SLS yields good results for CIFAR10 and CIFAR100, but performs poorly for ImageNet, as depicted in Figure 5. We attribute this outcome primarily to stability issues. Specifically, ADAM + SLS occasionally produces excessively large step sizes $\eta$, or it diminishes them to unreasonably small values $\eta \leq 10^{-10}$. On the other hand, our enhanced approaches ADAM + SaLSa and SGD + SaLSa, do not encounter these problems and on average deliver the best performance among all methods.

## 6 RELATED WORK

The optimization of deep neural networks has been a central topic of research in the field of machine learning. Various techniques and optimizers have been proposed, including but not limited to SGD (Robbins & Monro, 1951), Adagrad (Duchi et al., 2011), RADAM (Liu et al., 2020), ADAMW (Loshchilov & Hutter, 2019), RMSprop (Hinton & Swersky, 2014) and Adam (Kingma & Ba, 2015). However, selecting the most suitable optimizer remains a challenge, and there is no clear consensus on the best according to Schmidt et al. (2021).

Table 2: Final losses, averaged over 5 runs, for all datasets and optimization methods. Best performing (minimal loss) optimization method is marked in **bold**. The logarithmic average is taken due to the logarithmic nature of the typical loss.

|  | ADAM - | SGD - | ADAM SLS | SGD SLS | ADAM SaLSa | SGD SaLSa |
|---|---|---|---|---|---|---|
| *MNLI* | 0.009567 | 0.08613 | 0.03713 | 0.06901 | **0.005867** | 0.02174 |
| *QNLI* | 0.00258 | 0.02079 | 0.00504 | 0.03667 | **0.000628** | 0.0091627 |
| *MRPC* | 0.01312 | 0.1978 | 0.007298 | 0.05262 | **0.003126** | 0.03862 |
| *SST2* | **0.005857** | 0.02561 | 0.009457 | 0.0412 | 0.006991 | 0.01837 |
| GPT-2 | 2.86 | 3.572 | 2.917 | 3.566 | **2.772** | 3.559 |
| ResNet34 |  |  |  |  |  |  |
| *CIFAR10* | 0.01394 | 0.00982 | **0.0009508** | 0.05646 | 0.003314 | 0.003773 |
| *CIFAR100* | 0.03739 | 0.01143 | 0.01337 | 0.08245 | **0.003774** | 0.01453 |
| ResNet50 |  |  |  |  |  |  |
| *ImageNet* | 0.9122 | 1.547 | 2.036 | 1.144 | **0.8339** | 0.9788 |
| log average | 0.0355 | 0.0930 | 0.0315 | 0.134 | **0.0148** | 0.0477 |
| average rank | 2.75 | 4.625 | 3.125 | 5.5 | **1.25** | 3.75 |

In a recent study bySchmidt et al. (2021) on the topic of optimization methods, it was observed that while there are various optimizers available, there is no definitive best optimizer. The authors highlight that the introduction of more optimizers does not necessarily lead to improved results, and therefore, alternative approaches should be explored to enhance optimization techniques. One such approach with great potential is automatic step size determination. One of the most common approaches for this are line search methods, which hold promise for enhancing optimization processes (Nar & Sastry, 2018; Vaswani et al., 2019; Paquette & Scheinberg, 2020; Galli et al., 2023).

In this work we particularly build upon Vaswani et al. (2019). The Armijo line search method introduced there, offers several important advantages over other optimization techniques: no hyperparameter tuning of the learning rate, faster convergence rates and better generalization. A significant drawback of this method, along with other line search approaches, is that it requires at least an additional forward pass per update step. Consequently, this leads to an increase of approximately 30% in computational resources required per training step.

Recent work has shown that Transformers are highly sensitive to the choice of learning rate and learning rate schedule schedule during training Liu et al. (2020); Kenneweg et al. (2022). To address this issue, various approaches have been proposed, such as RADAM Liu et al. (2020) and warm starting. In this work, we show that our approach is able to train these highly sensitive architectures well. Other related work includes Granziol et al. (2022) which studies the correlation between batch size and learning rate, Streeter & Dillon (2022) which shows theoretical and practical results for training using higher order gradients, Arous et al. (2022) which studies the scaling limit of SGD in the high dimensional regime and Kunstner et al. (2023) which investigates why Adam is so effective at training the Transformer architecture.

The optimization of neural networks continues to be an important area of research, to which, the development of effective and reliable line search methods, which work on sensitive architectures such as transformers or large scale convolutional neural networks, constitutes a significant contribution.

## 7 CONCLUSION

We have introduced SaLSa, an automatic step size selection method and built a hyperparameter free general purpose optimizer on top. We have compared its performance against tuned learning rates for larger datasets and architectures than previously done in optimizer evaluations for line search methods. The SaLSa optimizer performance compares favorably in these cases, while requiring no tuning of learning rates, or code overhead, as well as minimal compute overhead. We recommend its use as a first choice for training deep neural networks in these domains and publish the code as a Python package.

## 8 REPRODUCIBILITY STATEMENT

We have taken great care to ensure reproducibility in this work. In the Appendix, we show all assumptions and the proof for Theorem 1. In the supplementary materials, we provide the exact code used for the experiments. We either described all pre-processing steps used in the experiments in Section 4 or referenced papers in which they are detailed. All datasets used are publicly available and are referenced accordingly.

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

# A APPENDIX

## A.1 BATCH SIZE SCALING

In a previous study by Granziol et al. (2022), the impact of batch size on the optimal step size was examined. Theoretical findings indicated that for optimization methods such as SGD, the optimal step size scales linearly with the batch size,tthods like ADAM or AMSGRAD, the scaling follows a square root relationship.

To investigate the behavior of the step size to batch size ratio in the ADAM + SaLSa optimizer, we conducted training runs with varying batch sizes. Remarkably, we observed that the step size to batch size ratio remained relatively constant throughout the training process. Moreover, the step size exhibited a scaling behavior of approximately $\eta \sim \sqrt{2}$, which aligns with the theoretically predicted optimal value described in Granziol et al. (2022). In Table 3, we present the average multiplicative factor by which the step size increased with respect to the batch size.

We take this as encouraging sign for the generalization abilities of our method.

The Armijo criterion 1 determines the step size by the ratio of loss decrease to gradient norm, visualized in 6. Notably, we observed that the primary factor influencing the reduction of step size is the increase in gradient norm, for lower batch sizes.

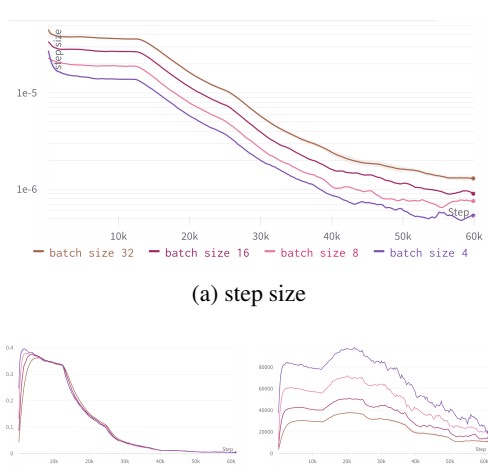

(a) step size

(b) loss decrease (c) gradient norm

Figure 6: Comparison of learning rates of ADAM + SaLSa dependent on the batch size on the MNLI dataset. Figure (b) depicts the loss decrease $h_k$ dependend on the batch size, here no great changes are present. Figure (c) depicts the gradient norm $s_k$ dependent on the batch size. Larger batches result in smaller gradient norms.

Table 3: Resulting average ratio of step size to previous step size of ADAM + SaLSa for doubling the batch size

| 4⇒8 | 8⇒16 | 16⇒32 | optimal from Granziol et al. (2022) |
|------|-------|--------|-------------------------------------|
| 1.325 | 1.423 | 1.420 | $\sqrt{2} \approx 1.414$ |

## A.2 ADDITIONAL EXPERIMENTAL RESULTS

Below we show additional accuracy and loss curves for the experiments.

Table 4: Ranking of classification accuracies, for all datasets and optimization methods. Best performing optimization method is marked in **bold**.

|  | ADAM - | SGD - | ADAM SLS | SGD SLS | ADAM SaLSa | SGD SaLSa |
|---|---|---|---|---|---|---|
| *MNLI* | 3 | 6 | **1** | 5 | 2 | 4 |
| *QNLI* | **1** | 5 | 3 | 6 | **1** | 4 |
| *MRPC* | 6 | 4 | 2 | 4 | **1** | 3 |
| *SST2* | **1** | 4 | 2 | 6 | 5 | 3 |
| ResNet34 |  |  |  |  |  |  |
| *CIFAR10* | 5 | 6 | 2 | **1** | 3 | 4 |
| *CIFAR100* | 6 | 4 | **1** | 5 | 2 | 3 |
| ResNet50 |  |  |  |  |  |  |
| *ImageNet* | 3 | 5 | 6 | 4 | **1** | 2 |
| average | 3.57 | 4.86 | 2.42 | 4.43 | **2.14** | 3.28 |

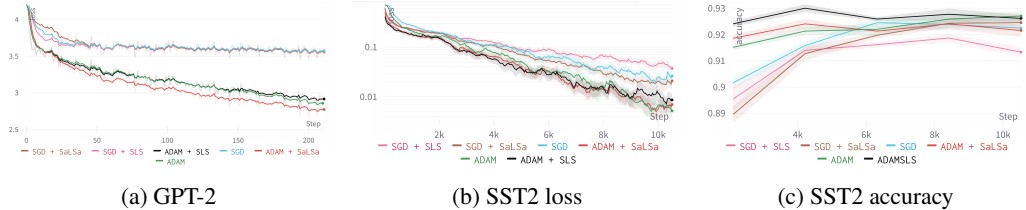

(a) GPT-2       (b) SST2 loss       (c) SST2 accuracy

Figure 7: The loss (top row) and accuracy curves of the EfficientNet experiment and the SST2 experiment and the GPT-2 experiment with standard error indicated around each line, starting after the first epoch. Accuracy was calculated on the validation data, while loss was calculated on the training data. No Accuracy is displayed for GPT-2, since it is not trained on a classification task.

Table 5: Ranking of final losses, for all datasets and optimization methods. Best performing optimization method is marked in **bold**.

|  | ADAM - | SGD - | ADAM SLS | SGD SLS | ADAM SaLSa | SGD SaLSa |
|---|---|---|---|---|---|---|
| *MNLI* | 2 | 6 | 4 | 5 | **1** | 3 |
| *QNLI* | 2 | 5 | 3 | 6 | **1** | 4 |
| *MRPC* | 3 | 4 | 2 | 6 | **1** | 5 |
| *SST2* | **1** | 5 | 3 | 6 | 2 | 4 |
| GPT-2 | 2 | 6 | 3 | 5 | **1** | 4 |
| ResNet34 |  |  |  |  |  |  |
| *CIFAR10* | 5 | 4 | **1** | 6 | 2 | 3 |
| *CIFAR100* | 5 | 2 | 3 | 6 | **1** | 4 |
| ResNet50 |  |  |  |  |  |  |
| *ImageNet* | 2 | 4 | 6 | 5 | **1** | 3 |
| average | 2.75 | 4.625 | 3.125 | 5.5 | **1.25** | 3.75 |

## A.3 PROOF FOR THEOREM 1

Below we show a theoretical proof for Theorem 1 and display training runs where we applied the non decrease condition $f(w_k) - f(w_k + \eta_k d_k) \geq 0$, by lowering the step size until it is fulfilled for each step, see Figure 10. The effect of this additional constraint does not affect the optimization process significantly.

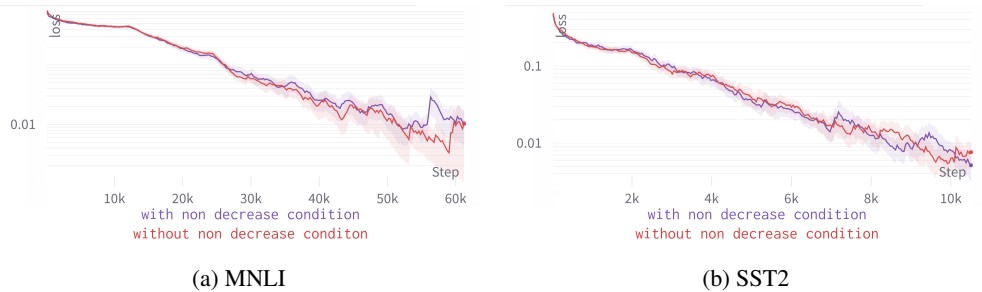

(a) MNLI          (b) SST2

Figure 8: Average Loss curves of training on the MNLI and SST2 dataset over 5 runs with and without the additional condition required in the proof.

*Proof.* The condition $f(w_k) - f(w_k + \eta_k d_k) \geq 0$ ensures that $\{f(w_k)\}_{k=1}^K$ is non-increasing and thus any infinite sequence will converge to $f(w^*)$, given the assumptions. It thus remains to show, that in every step such a step size $\eta_k$ can be found. By definition, we have that

$$h_k = \beta_3 h_{k-1} + (1 - \beta_3) [f(w_k) - f(w_k + \eta_k d_k)]$$
$$\geq \beta_3 c \eta_{k-1} s_{k-1} + (1 - \beta_3) [f(w_k) - f(w_k + \eta_k d_k)].$$

If we assume, that there exists a learning rate $\eta_k \leq \eta_{k-1}$, such that $f(w_k) - f(w_k + \eta_k d_k) \geq c\eta_k \|\nabla f(w_k)\|^2$ (see proof of existence below), we can show that

$$\beta_3 c \eta_{k-1} s_{k-1} + (1 - \beta_3) [f(w_k) - f(w_k + \eta_k d_k)] \geq c\eta_k \left[ \beta_3 s_{k-1} + (1 - \beta_3) \|\nabla f(w_k)\|^2 \right]$$
$$= c\eta_k s_k,$$

which finishes the proof, as we have found a learning rate $\eta_k$ that fulfills the SaLSa criterion.

We now prove the existence of $\eta_k \leq \eta_{k-1}$ with $f(w_k) - f(w_k + \eta_k d_k) \geq c\eta_k \|\nabla f(w_k)\|^2$ by contradiction, i.e. we assume that such a $\eta_k$ does not exist and thus $f(w_k) - f(w_k + \eta_k d_k) < c\eta_k \|\nabla f(w_k)\|^2$ for $\eta_k \leq \eta_{k-1}$. Using the Taylor expansion for $f$ around $f(w_k)$ yields

$$f(w_k) - f(w_k + \eta_k d_k) = -\eta_k d_k \nabla f(w_k) - o(\eta_k).$$

For $\eta_k \leq \eta_{k-1}$ it follows then

$$c\eta_k\|\nabla f(w_k)\|^2 > -\eta_k d_k \nabla f(w_k) - o(\eta_k).$$

Dividing both sides by $\eta_k$ and taking the limit for $\eta_k \to 0$ yields

$$c\|\nabla f(w_k)\|^2 > -d_k \nabla f(w_k),$$

and thus with $d_k = -\nabla f(w_k)$ it follows $c > 1$, which is a contradiction. $\qquad\square$

## A.4   HYPERPARAMETER STUDIES FOR $\beta_3$ AND $c$

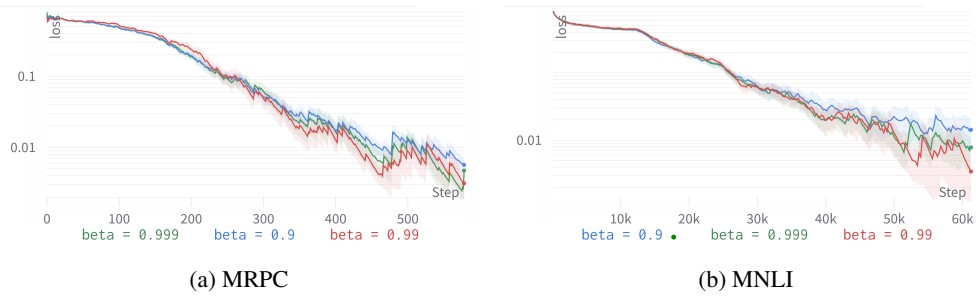

(a) MRPC                                      (b) MNLI

Figure 9: Average loss curves of training on the MRPC and SST2 dataset over 5 runs with different values for $\beta_3$ for the SaLSa + ADAM optimizer

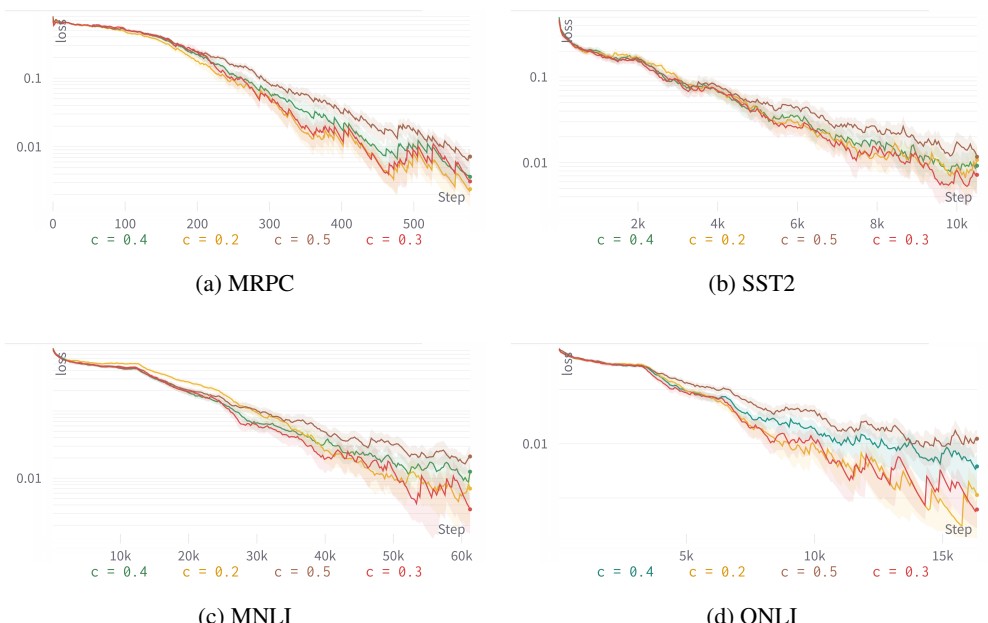

(a) MRPC                                      (b) SST2

(c) MNLI                                       (d) QNLI

Figure 10: Average loss curves with standard error indicated of training on the GLUE dataset over 5 runs with different values for $c$ for the SaLSa + ADAM optimizer

In our Hyperparameter studies, we notice that the $\beta_3$ parameter minimally affects performance, with all training runs falling within the range of estimation error. The hyperarameter $c$ however does have a more substantial impact on the performance. All runs converge regardless of tested value, yet higher $c$ values exhibit a tendency to converge at a slower pace overall.

## A.5   LIMITING THE LINE SEARCH FREQUENCY

Here we show experimental results evaluating the impact of the proposed speed-up in Section 3.4.
The observed difference in performance in Figure 11 is minor and within the margin of error.

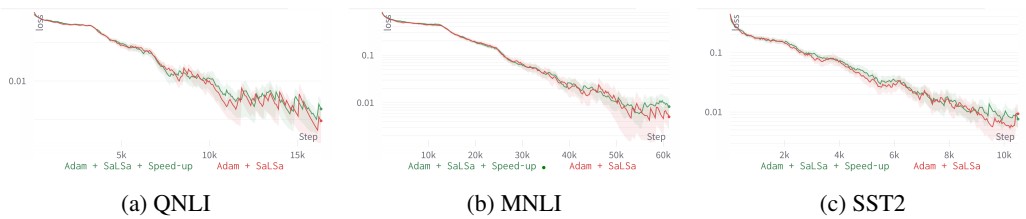

(a) QNLI                    (b) MNLI                    (c) SST2

Figure 11: Average loss curves with standard error indicated of training on the SST2, MNLI and QNLI dataset over 5 runs with and without the speed-up for the SaLSa + ADAM optimizer

In 12 we can observe that $L_k$ is maximal $L_k = 10$ during long plateaus of the step size, but decreases for faster changes in the step size.

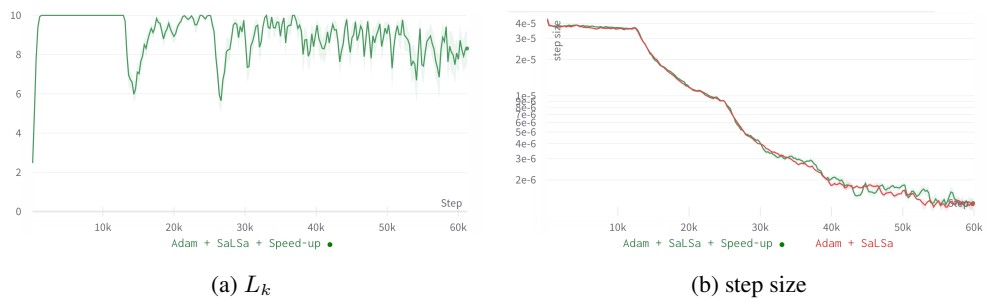

(a) $L_k$                         (b) step size

Figure 12: $L_k$ and step size curves with standard error indicated of training on the MNLI dataset over 5 runs with the SaLSa + ADAM + Speed-up optimizer

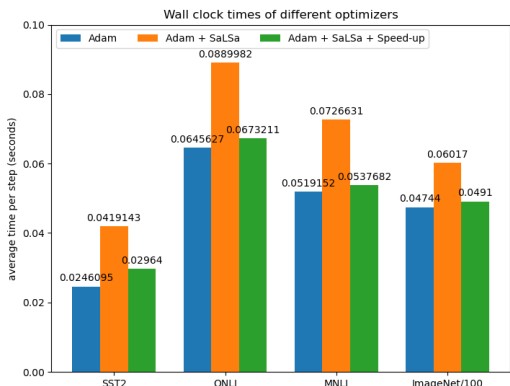

Figure 13: Wall clock running times for different optimization methods and datasets. All experiments done on a single A40 GPU. ImageNet times are scaled down by a factor of 100.

