# OpenReview forum: "No learning rates needed: Introducing SaLSa - Stable Armijo Line Search Adaptation"
_ICLR.cc/2024/Conference — Submitted to ICLR 2024_

### Official Review · Reviewer_9vmP · 2023-10-31

**Soundness:** 3 good
**Presentation:** 3 good
**Contribution:** 3 good
**Rating:** 8
**Confidence:** 5

**Summary:**

This paper proposes an exponentially smoothed Armijo condition to improve the stability of line-search algorithms for finding the step size of Adam. Specifically, it builds upon the preconditioned gradient search condition proposed by Vaswani et al. [2], and applies exponential moving averages to both sides of the condition to improve the stability and robustness of the found step size to mini-batch noises and random parameter initializations. Experiments are performed on both image and NLP tasks to verify the effectiveness of the proposed method.

**Strengths:**

- This paper addresses an important problem, i.e., the sensitivity of the found step size to mini-batch noises, in stochastic line-search algorithms[1, 2]. The experiments seem to be comprehensive.

- I have tested the algorithm. The algorithm seems to fix the problem of learning rate drop observed in both SLS [1] and its variants for adaptive step sizes [2]. The resulting performance of the proposed algorithm indeed improves upon previous works.

[1] Vaswani et al. Painless Stochastic Gradient: Interpolation, Line-Search, and Convergence Rates

[2] Vaswani et al. Adaptive Gradient Methods Converge Faster with Over-Parameterization (but you should do a line-search)

**Weaknesses:**

- The theoretical guarantee is very weak. The assumption that the loss is monotonically decreasing is not realistic given the actual update direction includes momentum. This is further complicated by the fact that the step size is found by the modified (smoothed) Armijo condition rather than the preconditioned gradient.

- I am a little bit suspicious that the step size drop in Figure 3 is caused by numerical issues. In fact, the sudden drop may happen across several iterations caused by mini-batch noises when the gradient norm becomes small. The authors may want to comment more on this.

**Questions:**

In a more recent work [3], the (stochastic) Armijo condition has been relaxed to allow the found step size not guarantee descent. Such non-monotone condition seems to fix the learning rate drop problem. The authors may want to consider adding this work to their baselines. Despite the theoretical weakness, I think this work is significant and would like to accept it.

[3] Galli et al. Don’t be so Monotone: RelaxingStochastic Li neSearch in Over-Parameterized Models

---

> ### Author Response · Authors · 2023-11-21
>
> Thanks for the very in depth review and your dedication in testing the algorithm.
>
>
> Weakness:
>
> > The theoretical guarantee is very weak. The assumption that the loss is monotonically decreasing is not realistic given the actual update direction includes momentum. This is further complicated by the fact that the step size is found by the modified (smoothed) Armijo condition rather than the preconditioned gradient.
>
>
> A: You are correct in pointing out that this is not a particularly strong theoretical result, however we feel that it is still important to show that we can at least provide some theoretical guarantees. This is not possible for a general case even for the ADAM optimizer alone as momentum terms in general do not allow for convergence guarantees for arbitrary loss functions.
>
>
>
>
> > I am a little bit suspicious that the step size drop in Figure 3 is caused by numerical issues. In fact, the sudden drop may happen across several iterations caused by mini-batch noises when the gradient norm becomes small. The authors may want to comment more on this.
>
>
> A: We are confident that some of the sudden drop stems from numerical issues since the gradient norm drops to true zero, which is not mathematically possible for a non-zero gradient, which we did not observe. We leave a more thorough analysis of this phenomenon for future work.
>
>
> Questions:
>
> Q: In a more recent work [3], the (stochastic) Armijo condition has been relaxed to allow the found step size not guarantee descent. Such non-monotone condition seems to fix the learning rate drop problem. The authors may want to consider adding this work to their baselines.
>
> A: This is definitely an interesting work. We will consider adding this work as another comparison point in future versions of this paper and added it to the related work.

---

> ### Comment · Reviewer_9vmP · 2023-12-01
> **Reply**
>
> Thanks for the rebuttal. My score stays the same.

---

### Official Review · Reviewer_XiSh · 2023-10-31

**Soundness:** 2 fair
**Presentation:** 3 good
**Contribution:** 2 fair
**Rating:** 3
**Confidence:** 5

**Summary:**

The paper modifies the Armijo line search method incorporating a momentum term into the Armijo criterion.

**Strengths:**

1. the paper start with interesting practical observations of existing method, and improve it via simple approach.
2. the paper is well written and easy to follow.

**Weaknesses:**

I do not see much benefit of the proposed method over the current lr schedule. As a practitioner, I do not have a strong motivation to use the proposed method after reading the paper. The reasons are as follows:

1. **Fine-tuning tasks are too simple, not convincing.**  Most NLP experiments are based on small fine-tuning datasets and small models.  These tasks are simple for optimization and insensitive to LR choices.  I am hoping to see more challenging tasks such as pretraining tasks for larger models on larger datasets, which rely much heavier on a good lr choice than the fine-tuning tasks shown in the script.
2.  **Unsatisfactory performance.**   The performance on imagenet tasks is unsatisfactory. Unclear performance on pretraining tasks on larger datasets and larger models.
3. **Did not save the number of hyperparameters.**  The proposed method has many hyperparameters. For instance, there are at least three hyperparameters in section 2.1 (including initial lr,  c, delta, b ), beta3 in section 3.1, and two betas in section 3.4.  As such, the proposed method does not save much trouble in the current lr schedule. The title "No learning rate needed" is overclaimed, as well.
4. **More runing time**.  Backtracking methods require additional forward and backward passing of neural nets, which could be the major computational bottleneck when the model size grows. Though the authors claim they only require 3% additional running time over standard training,  there is no real evidence to support this claim.  I would suggest the authors report the wall-clock running time and compare it to that of standard training. Further, as the model size grows, the additional forward & backward passing would require much more running time as claimed 3%.



In summary, I don't see the motivation to replace the current lr schedule with the proposed method.

**Questions:**

1. In Eq.2: Is v a vector or a scaler?
2. Wrong template? The current script is using ICLR 2023 template, instead of ICLR 2024

---

> ### Author Response · Authors · 2023-11-21
>
> Thanks for your valuable feedback. We answer point by point.
>
> Weakness:
>
>
> > Fine-tuning tasks are too simple, not convincing. Most NLP experiments are based on small fine-tuning datasets and small models. These tasks are simple for optimization and insensitive to LR choices. I am hoping to see more challenging tasks such as pretraining tasks for larger models on larger datasets, which rely much heavier on a good lr choice than the fine-tuning tasks shown in the script.
>
>
>
> A: To address the suitability of SalSa for pretraining tasks we present full pre-training on Image tasks in which our algorithm outperforms the existing baselines. Furthermore, we argue that fine tuning tasks are the most common use case for a practitioner, since pre-training is often infeasible due to resource limitations. Further, it can empirically be observed that these tasks are also sensitive to the learning rate, especially larger sized fine tuning tasks on datasets like MNLI.
>
>
> > Unsatisfactory performance. The performance on imagenet tasks is unsatisfactory. Unclear performance on pretraining tasks on larger datasets and larger models.
>
>
> A: The original ResNet paper on ImageNet trained for 12 epochs instead of our 5, used additional data augmentation and performed 10 crop testing to obtain their final score. We omit additional data augmentation and 10 crop testing, since we are not interested in peak performance but rather the comparative performance between different optimizers. To be more comparable to the original ResNet paper we have increased our training time to 12 epochs, which resulted in the same relative results of the training schemes.
>
>
>
> > Did not save the number of hyperparameters. The proposed method has many hyperparameters. For instance, there are at least three hyperparameters in section 2.1 (including initial lr, c, delta, b ), beta3 in section 3.1, and two betas in section 3.4. As such, the proposed method does not save much trouble in the current lr schedule. The title "No learning rate needed" is overclaimed, as well.
>
>
> A: While we do not technically save on the number of hyperparameters, a default choice of these hyperparameters for all training tasks is sufficient. In the ablation studies we have shown extensively that SaLSa is very robust to these hyperparameters. In practice a user does not need to adapt these parameters to accommodate their task.
>
>
> > More running time. Backtracking methods require additional forward and backward passing of neural nets, which could be the major computational bottleneck when the model size grows. Though the authors claim they only require 3% additional running time over standard training, there is no real evidence to support this claim. I would suggest the authors report the wall-clock running time and compare it to that of standard training. Further, as the model size grows, the additional forward & backward passing would require much more running time as claimed 3%.
>
> A: To provide further evidence for our estimated compute we added a wall-clock running time table in the appendix, see Figure 13. There we can see that especially for larger datasets (for example ImageNet, MNLI) 3% is a valid estimate. For smaller datasets (QNLI, SST2) initialization and other overhead results in a slightly higher percentage of extra compute.
> In the paper we explain that our line search takes on average 30% extra compute per step, since it needs on average about 1 extra forward pass, no extra backward pass is ever needed. We only have to perform our line search on average every 10 steps.
>
> Questions:
>
>
> Q: In Eq.2: Is v a vector or a scaler?
>
> A: The “v” in Equation 3 is a vector.
>
>
> Q: Wrong template? The current script is using ICLR 2023 template, instead of ICLR 2024
>
> A: Thanks for pointing this out, we changed it to the ICLR 2024 template.

---

> > ### Comment · Reviewer_XiSh · 2023-11-23
> > **Thanks for the response, I will keep my score**
> >
> > Thanks for the response, I will keep my score for the following reasons:
> >
> > 1. Fine-tuning tasks on small or medium-sized model (e.g., Bert) is too simple. Shakespeare dataset is too toy. There is no sufficient evidence to support the efficacy of the proposed algorithm.
> >
> > 2. The proposed method did not save the amount of hyperparameters. The title "No learning rate needed" is overclaimed. The authors argue that "SaLSa is very robust to these hyperparameters", but this might be due to the reason that the experimental settings are intrinsically simple and insensitive to the hyperparameters.

---

### Official Review · Reviewer_F4Yh · 2023-11-01

**Soundness:** 3 good
**Presentation:** 4 excellent
**Contribution:** 3 good
**Rating:** 6
**Confidence:** 3

**Summary:**

This work presents a modification to Stochastic Line Search for online learning rate adaptation, which 1) introduce momentum to the line search parameters and 2) gates the frequency of the execution of the line search. The authors motivate these additions with a thorough explication of relevant background, including theoretical analysis, intuitive justification, and concrete demonstration of the limitations of the original method. The authors present thorough experiments across image and language tasks, with significant variance in model architecture and dataset size. The results show impressive average performance over baselines of ADAM/SGD with and without line search. The authors release the code for public use.

**Strengths:**

The paper is clearly written. The motivation and theoretical analysis are clear. The thoroughness of the experimental setup is excellent. The contribution is very significant and I think likely to be very useful to machine learning practitioners in general.

**Weaknesses:**

While the presented results are very compelling, it would be even stronger to show experiments on Transformer models in a pretaining setting as well. Additionally, it would be beneficial to compare different sizes of Transformer (and CNN) to show the persistence of the benefit of the method across the scaling of the model. These experiments may plausibly be left as future work (the presented ablations are more important and very thorough) but they represent practical questions that ML practitioners will have, so would strengthen the current work if included.

**Questions:**

Nits:
Figs 4 and 5: it is a little difficult to read these plots, as you have to go back and forth visually between the plot and the legend to see which curve is which. Consider making Salsa lines dashed / SLS lines dotted / baselines solid (or something of that nature) to help improve readability.
Fig 5: loss label missing at top of (a), accuracy label missing at top of (d)
Fig 1: fix occluded label text for y axis

---

> ### Author Response · Authors · 2023-11-21
>
> Thank you for this excellent review
>
>
> Weaknesses:
>
>
> Yes this would be great, we leave this to future work.
>
>
> Questions:
>
>
> We incorporated all the mentioned suggestions for readability, with the exception of dashed/dotted lines. Thanks for pointing this out.

---

### Official Review · Reviewer_QdFA · 2023-11-01

**Soundness:** 2 fair
**Presentation:** 2 fair
**Contribution:** 3 good
**Rating:** 3
**Confidence:** 3

**Summary:**

The paper presents SaLSa, an approach to improving Armijo line search by incorporating a momentum term to better handle stochastic mini-batches. This is shown to outperform results without use of the buffer on a variety of different datasets.

**Strengths:**

- The proposed approach of applying smoothing to deal with noise from sampling mini-batches is intuitively and theoretically justified.
- The authors present results demonstrating competitive performance on a couple of different domains.

**Weaknesses:**

- I'm not too familiar with the previous SLS literature, but it does not seem like a particularly surprising result that a smoothed version of the algorithm, together with a guarantee that every learning rate yields an improved loss results in convergence.
- I'm a bit concerned by the quality of the baselines. The accuracies achieved on ImageNet with vanilla SGD seem abnormally low; more care should be used in making sure the baselines ("For the image tasks we compare to a flat learning rate" seems like an unfair comparison to the baseline since line search is able to adapt learning rates. It would be better to compare to the commonly used waterfall schedule or linear decay schedule)
- "Peak classification accuracy" does not seem to be a standard metric -- it would be preferable to include the accuracy at the end of training or the test accuracy at the checkpoint with the highest validation accuracy.
- The writing is in general a bit loose and motivation for design choices could be strengthened. There are some issues with spelling and grammar.

**Questions:**

- In general, need to be more careful in using \citep or \citet
- It would be helpful to show that the problem depicted in Figures 1 and 3 is alleviated by using SaLSa.
- It's a bit strange to report the average or log average final loss as an aggregate metric, given how different the models and datasets are. Perhaps it would be more informative to report the relative ranks of each method.
- There are cases where SLS outperforms SaLSa -- it would be helpful to investigate further to understand why.

Minor
- full-filed -> fulfilled
- having an average -> have an average

---

> ### Author Response · Authors · 2023-11-21
>
> Thanks for your valuable feedback. We answer point by point.
>
> Questions:
>
> Q: In general, need to be more careful in using \citep or \citet
>
> A: Thanks for pointing this out we fixed it according to the ICLR 2024 guidelines
>
>
> Q: It would be helpful to show that the problem depicted in Figures 1 and 3 is alleviated by using SaLSa.
>
> A: We added SaLSa step sizes to Figures 1 and 3 to show how SaLSa does fix the problem.
>
>
> Q: It's a bit strange to report the average or log average final loss as an aggregate metric, given how different the models and datasets are. Perhaps it would be more informative to report the relative ranks of each method.
>
> A: We added the relative ranks in Tables 4 and 5 in the appendix and added the average rank at the bottom of Table 1 and 2
>
>
> Q: There are cases where SLS outperforms SaLSa -- it would be helpful to investigate further to understand why.
>
> A: This is definitely interesting, but we leave it for future work.
>
> Weakness:
> > I'm not too familiar with the previous SLS literature, but it does not seem like a particularly surprising result that a smoothed version of the algorithm, together with a guarantee that every learning rate yields an improved loss results in convergence.
>
>
> A: Note that an analogous convergence result is not possible for the general case (even for the unaugmented ADAM optimizer) as momentum terms in general do not allow for convergence guarantees for arbitrary loss functions. Hence we find it important to specify relevant restrictions in which we can provide theoretical guarantees.
>
>
> > I'm a bit concerned by the quality of the baselines. The accuracies achieved on ImageNet with vanilla SGD seem abnormally low; more care should be used in making sure the baselines ("For the image tasks we compare to a flat learning rate" seems like an unfair comparison to the baseline since line search is able to adapt learning rates. It would be better to compare to the commonly used waterfall schedule or linear decay schedule)
>
>
> A:
> The original ResNet paper on ImageNet trained for 12 epochs instead of 5, used additional data augmentation and performed 10 crop testing to obtain their final score. We omit additional data augmentation and 10 crop testing, since we are not interested in peak performance but rather the comparative performance between different optimizers. For better comparison to the original ResNet paper we have increase training time to 12 epochs, resulting in better results - yet the relative performance of the optimizers did not change.
>
> With regards to more complex learning rate schedules: We showed a more complex learning rate schedule as a stronger baseline, specifically the cosine decay schedule for the NLP experiments. For the image datasets we used a flat baseline to allow comparability to the experiments from previous SLS papers [1,2].
>
>
> > "Peak classification accuracy" does not seem to be a standard metric -- it would be preferable to include the accuracy at the end of training or the test accuracy at the checkpoint with the highest validation accuracy.
>
> A: I think this is a misunderstanding. We did not perform any hyperparameter tuning on a single dataset for our methods and thus did not use a train, validation and test split. Consequently, our validation and test sets are the same and “peak classification accuracy” is the accuracy at the checkpoint with the highest validation accuracy.
>
>
>
>
> [1] Sharan Vaswani, Aaron Mishkin, Issam Laradji, Mark Schmidt, Gauthier Gidel, and Simon Lacoste-Julien. Painless stochastic gradient: Interpolation, line-search, and convergence rates.NIPS’19:Proceedings of the 33rd International Conference on Neural Information Processing Systems,2019.
>
>
> [2] Sharan Vaswani, Issam H. Laradji, Frederik Kunstner, Si Yi Meng, Mark Schmidt, and Simon Lacoste-Julien. Adaptive gradient methods converge faster with over-parameterization (and you can do aline-search). 2021.

---

> ### Comment · Reviewer_QdFA · 2023-11-22
> **Response/ Some additional questions**
>
> Thanks for the response! Some additional thoughts/response:
>
> > For the image datasets we used a flat baseline to allow comparability to the experiments from previous SLS papers [1,2].
>
> I believe it is still valuable to use a decay, as a practitioner would not use a constant learning rate schedule (and this would be a shortcoming of the prior work as well). Additionally, training for a shorter horizon can result in different algorithm performance that may be suboptimal for longer horizons e.g. "Understanding Short-Horizon Bias" from Yuhuai Wu et al.
>
> > We did not perform any hyperparameter tuning on a single dataset
>
> To clarify, does that mean you performed hyperparameter tuning across all datasets? e.g. the paragraph says "In our experiments we found good values for c to be in the range c ∈ [0.3, 0.5]. For all our experiments we used c = 0.3
> (compared to c = 0.1 for the original Armijo line search criterion...Furthermore, we tuned the hyperparameter β3 ∈ [0.9, 0.999] from Eq. 7 on a variety of datasets."
>
> Does this lead to an unfair comparison if you spent more compute on tuning your approach?
>
> My original point is peak classification accuracy is generally a poor metric because we do not in general have access to the true test set and is thus not possible to measure. It would also reward a method that is high variance and occasionally finds good solutions.

---

> ### Author Response · Authors · 2023-11-23
>
> >  Additionally, training for a shorter horizon can result in different algorithm performance that may be suboptimal for longer horizons e.g. "Understanding Short-Horizon Bias" from Yuhuai Wu et al.
>
> We agree with this sentiment which is why we increased the training length from 5 to 12 epochs after your feedback to be the same as in the original ResNet Paper. The relative performance however did not change.
>
> > Does this lead to an unfair comparison if you spent more compute on tuning your approach?
>
> We performed a grid search for the optimal learning rate for ADAM and SGD for the image datasets, which used a comparatively large amount of compute. In contrast for our method we performed the hyperparameter tuning only once on the datasets indicated in the Appendix but not for all datasets. We also found that in contrast to the learning rate our hyperparameters are relatively insensitive to change, especially $\beta_3$.
>
> > My original point is peak classification accuracy is generally a poor metric because we do not in general have access to the true test set and is thus not possible to measure. It would also reward a method that is high variance and occasionally finds good solutions.
>
> We observe that for most datasets we tested, the peak and final accuracy are the same. We also show in Figures 4 and 5 that our methods do not exhibit very high variance, but rather consistently find good solutions.

---

### Meta-Review · Area_Chair_rjjA · 2023-12-02

**Metareview:**

The paper proposes an improvement to the Armijo line search technique by adding a momentum term to the Armijo criterion. Overall, the paper addresses an important research direction, considering the importance of quickly converging optimization techniques for Deep Learning. However, the reviewers and myself think that the paper is not ready for publication in its current form. In particular, the experiments in the paper are insufficient because they involve small or medium-sized models on fine-tuning tasks, where the impact of learning rate schedules might be less apparent. Furthermore, the proposed paper should better ablate the impact of its hyperparameters, since the hyperparameterless claim of the technique is not thoroughly validated (the method introduces its hyperparameters, whose impact needs to be further investigated). While the paper is not ready for publication in its current form, I advise the authors to conduct experiments on large-scale models to demonstrate the relevance of the method for the Deep Learning community.

**Justification For Why Not Higher Score:**

The experiments are not convincing.

**Justification For Why Not Lower Score:**

N/A

---

### Decision · Program_Chairs · 2024-01-16

Reject